# Cardiometabolic risk factors in South American children: A systematic review and meta-analysis

**Carolyn M. H. Singleton**[1], **Sumeer Brar**[1], **Nicole Robertson**[1], **Lauren DiTommaso**[1], **George J. Fuchs, III**[2,3], **Aric Schadler**[4], **Aurelia Radulescu**[4], **Suzanna L. Attia**[2]*

**1** University of Kentucky College of Medicine, Lexington, Kentucky, United States of America, **2** University of Kentucky College of Medicine Division of Pediatric Gastroenterology, Hepatology and Nutrition, Kentucky, United States of America, **3** University of Kentucky College of Public Health Department of Epidemiology, Kentucky, United States of America, **4** University of Kentucky College of Medicine Department of Pediatrics, Kentucky, United States of America

* suzango@gmail.com

## Abstract

### Background

Cardiometabolic risk factors (impaired fasting glucose, abdominal obesity, high blood pressure, dyslipidemia) cluster in children, may predict adult disease burden, and are inadequately characterized in South American children.

### Objectives

To quantify the burden of cardiometabolic risk factors in South American children (0–21 years) and identify knowledge gaps.

### Methods

We systematically searched PubMed, Google Scholar, and the Latin American and Caribbean Health Sciences Literature via Virtual Health Library from 2000–2021 in any language. Two independent reviewers screened and extracted all data.

### Results

179 studies of 2,181 screened were included representing 10 countries (n = 2,975,261). 12.2% of South American children experienced obesity, 21.9% elevated waist circumference, 3.0% elevated fasting glucose, 18.1% high triglycerides, 29.6% low HDL cholesterol, and 8.6% high blood pressure. Cardiometabolic risk factor definitions varied widely. Chile exhibited the highest prevalence of obesity/overweight, low HDL, and impaired fasting glucose. Ecuador exhibited the highest prevalence of elevated blood pressure. Rural setting (vs. urban or mixed) and indigenous origin protected against most cardiometabolic risk factors.

**Data Availability Statement:** All articles included in our meta-analysis are publicly available and are listed in our article.

**Funding:** The authors received no specific funding for this work.

**Competing interests:** The authors declare that no competing interests exist.

**Abbreviations:** CMRF, Cardiometabolic risk factors; HBP, High blood pressure; MetS, Metabolic syndrome; IDF, International Diabetes Foundation; AHA, American Heart Association; BMI, Body mass index; WC, Waist circumference; GI, Glucose Intolerance; HDL, high-density lipoprotein; LDL, Low-density lipoprotein; WHO, World Health Organization.

## Conclusions

South American children experience high rates of obesity, overweight, and dyslipidemia. International consensus on cardiometabolic risk factor definitions for children will lead to improved diagnosis of cardiometabolic risk factors in this population, and future research should ensure inclusion of unreported countries and increased representation of indigenous populations.

## Introduction

Metabolic syndrome is the clustering of the cardiometabolic risk factors (CMRF) of impaired fasting glucose, abdominal obesity, high blood pressure (HBP), and dyslipidemia [1]. As of 2017, 20–25% of the world's population was estimated to have metabolic syndrome (MetS) with an associated two or three times increased risk of death by heart attack or stroke respectively [1]. Childhood obesity is on the rise globally with an inconsistent understanding of the burden and impact of MetS in childhood and beyond [2]. Children with obesity are more likely to have obesity in adulthood, which is itself associated with significant health complications including gallstones, type 2 diabetes mellitus, non-alcoholic fatty liver disease, osteoarthritis, certain cancers, and cardiovascular events [3]. A rapid rise in obesity over the past fifty years is associated with a rise of 70% in healthcare treatment costs, and this now outweighs spending on non-communicable diseases associated with tobacco use or alcohol dependence [3].

### Defining cardiometabolic risk factors and metabolic syndrome in children

More than 40 different definitions of MetS in the pediatric population exist [4]. Two notable attempts to approach MetS definition in the pediatric population have been made by the International Diabetes Foundation (IDF) and the American Heart Association (AHA) [4]. The IDF definition states that the adult definition can be used down to the age of 10 years with the exception of a change in the definition of abdominal obesity, which is defined in children as ≥90th percentile for age and gender. The AHA did not explicitly create their own guidelines, but rather pointed to the conclusions of three studies, Cook et al, de Ferranti et al, and Ford et al, as appropriate starting points [4–7].

The rise in global obesity and CMRF is particularly relevant in South American countries, where cardiovascular disease is now the most common cause of death and disability [8]. Despite the increasing attention on CMRF globally and their short- and long-term effects both in populations and in an individual's lifespan, the burden of CMRF in South American children is poorly defined [9]. Filling this knowledge gap could contribute to reducing this disease burden by illuminating the most common CMRF and bringing policy and healthcare attention to diagnosis, prevention, and treatment of at-risk children.

### Objective

The objective of this systematic review and meta-analysis was to quantify the burden of cardiometabolic risk factors in South American children, identify knowledge gaps, and propose next steps for research.

## Methods

### Inclusion/Exclusion criteria

**Inclusion Criteria:** We included all primary quantitative data from human subject studies on children ages 0–21 years from the geographical region of South America reported in any language. We included studies that reported prevalence data using any definition of at least one of the following: glucose intolerance; obesity; elevated waist circumference (WC); high blood pressure; and/or dyslipidemia, considered as low high-density lipoprotein (HDL), high low-density lipoprotein (LDL), and/or high triglyceride (TG). **Exclusion Criteria:** We excluded studies without available data, with data collected before 2000, focusing on a population with a chronic and/or congenital medical condition, duplicate data across multiple publications, duplicate data reported in different languages, and that did not report cutoff definitions for CMRF. We excluded the Caribbean islands due to their increased cultural and culinary heterogeneity compared to South America. Studies who recruited participants solely based on overweight/obese status were also excluded in order to minimize selection bias, as CMRF are higher in obese populations.

### Search strategy

Our search strategy is detailed in Table 1. We searched all available medical literature, including gray literature, using keyword searches in relevant databases. Our search strategy was composed of three main steps: 1) we performed an initial search of primary literature aggregators, ex. PubMed, to determine appropriate keyword terms in titles and abstracts; 2) we then used appropriate keywords to conduct a more thorough search of the literature; finally, 3) we examined references of articles found in step two to identify additional relevant data sources. We used Preferred Reporting Items for Systematic Review and Meta-analyses Protocols (PRISMA-P) as a framework to develop and guide our review [10]. An electronic search was performed using the databases PubMed, the Latin American and Caribbean Health Sciences Literature via Virtual Health Library, and Google Scholar. We examined the grey literature through the Google Scholar search. The initial search terms included: *"Cardiometabolic AND risk factor AND Latin America NOT adult," "Cardiometabolic risk factor Ecuador,"* and *"Metabolic syndrome [MeSH Term] AND Latin America [MeSH Term]."* MeSH terms were used when searching PubMed. The keywords around which the searches were developed included phrases revolving around CMRF, children, adolescents, and South America or Ecuador. We used Ecuador as the initial country to test and develop our search terms. This search strategy, specifically in Google Scholar, did not adequately represent all South American countries. Therefore, we expanded the Google Scholar search with advanced search features to combine all terms for each CMRF and individually searched all spelling variations for each South American country plus "children". Fig 1 shows the PRISMA flow chart and screening process of all articles for the first search strategy.

PRISMA 2009 flow diagram. From: Moher D, Liberati A, Tetzlaff J, Altman DG, The PRISMA Group (2009). Preferred Reporting Items for Systematic Reviews and Meta-Analyses: The PRISMA Statement. PLoS Med [10].

### Study selection and data extraction

One reviewer (CH) screened titles and abstracts of the retrieved articles for relevance, and a second reviewer (SB) independently confirmed relevance. Disputes were settled by an arbitrator (SLA). Fig 1 shows the associated PRISMA flowchart of study selection. Two independent people (CH, SB, LD, and/or NR) performed data extraction and assessment of study quality.

**Table 1. Search strategy.**

| Database | Search Terms | Hits |
|---|---|---|
| PubMed | (cardiometabolic risk children adolescents South America) OR (cardiometabolic risk factor and "Latin America"[Mesh]) | 74 |
| Latin American and Caribbean Health Sciences Literature via Virtual Health Library | cardiometabolic risk factors children | 43 |
| Google Scholar Basic Search | cardiometabolic risk factors in Ecuadorian children | 1,060* |
| Google Scholar Advanced Searches | allintitle: Argentina children obesity OR pressure OR systolic OR diastolic OR SBP OR DBP OR HBP OR HDL OR LDL OR TG OR hyperinsulinemia OR insulin OR "waist circumference" OR overweight OR anthropomorphic OR dyslipidemia OR glucose OR lipid OR cholesterol | 21 |
| | allintitle: Argentinian children obesity OR pressure OR systolic OR diastolic OR SBP OR DBP OR HBP OR HDL OR LDL OR TG OR hyperinsulinemia OR insulin OR "waist circumference" OR overweight OR anthropomorphic OR dyslipidemia OR glucose OR lipid OR cholesterol | 5 |
| | allintitle: Argentinean children obesity OR pressure OR systolic OR diastolic OR SBP OR DBP OR HBP OR HDL OR LDL OR TG OR hyperinsulinemia OR insulin OR "waist circumference" OR overweight OR anthropomorphic OR dyslipidemia OR glucose OR lipid OR cholesterol | 22 |
| | allintitle: Bolivia children obesity OR pressure OR systolic OR diastolic OR SBP OR DBP OR HBP OR HDL OR LDL OR TG OR hyperinsulinemia OR insulin OR "waist circumference" OR overweight OR anthropomorphic OR dyslipidemia OR glucose OR lipid OR cholesterol | 3 |
| | allintitle: Bolivian children obesity OR pressure OR systolic OR diastolic OR SBP OR DBP OR HBP OR HDL OR LDL OR TG OR hyperinsulinemia OR insulin OR "waist circumference" OR overweight OR anthropomorphic OR dyslipidemia OR glucose OR lipid OR cholesterol | 4 |
| | allintitle: Brazil children obesity OR pressure OR systolic OR diastolic OR SBP OR DBP OR HBP OR HDL OR LDL OR TG OR hyperinsulinemia OR insulin OR "waist circumference" OR overweight OR anthropomorphic OR dyslipidemia OR glucose OR lipid OR cholesterol | 103 |
| | allintitle: Brazilian children obesity OR pressure OR systolic OR diastolic OR SBP OR DBP OR HBP OR HDL OR LDL OR TG OR hyperinsulinemia OR insulin OR "waist circumference" OR overweight OR anthropomorphic OR dyslipidemia OR glucose OR lipid OR cholesterol | 95 |
| | allintitle: Brasil children obesity OR pressure OR systolic OR diastolic OR SBP OR DBP OR HBP OR HDL OR LDL OR TG OR hyperinsulinemia OR insulin OR "waist circumference" OR overweight OR anthropomorphic OR dyslipidemia OR glucose OR lipid OR cholesterol | 7 |
| | allintitle: Brasilian children obesity OR pressure OR systolic OR diastolic OR SBP OR DBP OR HBP OR HDL OR LDL OR TG OR hyperinsulinemia OR insulin OR "waist circumference" OR overweight OR anthropomorphic OR dyslipidemia OR glucose OR lipid OR cholesterol | 0 |
| | allintitle: Chile children obesity OR pressure OR systolic OR diastolic OR SBP OR DBP OR HBP OR HDL OR LDL OR TG OR hyperinsulinemia OR insulin OR "waist circumference" OR overweight OR anthropomorphic OR dyslipidemia OR glucose OR lipid OR cholesterol | 30 |
| | allintitle: Chilean children obesity OR pressure OR systolic OR diastolic OR SBP OR DBP OR HBP OR HDL OR LDL OR TG OR hyperinsulinemia OR insulin OR "waist circumference" OR overweight OR anthropomorphic OR dyslipidemia OR glucose OR lipid OR cholesterol | 65 |
| | allintitle: Colombia children obesity OR pressure OR systolic OR diastolic OR SBP OR DBP OR HBP OR HDL OR LDL OR TG OR hyperinsulinemia OR insulin OR "waist circumference" OR overweight OR anthropomorphic OR dyslipidemia OR glucose OR lipid OR cholesterol | 21 |
| | allintitle: Colombian children obesity OR pressure OR systolic OR diastolic OR SBP OR DBP OR HBP OR HDL OR LDL OR TG OR hyperinsulinemia OR insulin OR "waist circumference" OR overweight OR anthropomorphic OR dyslipidemia OR glucose OR lipid OR cholesterol | 23 |
| | allintitle: Ecuador children obesity OR pressure OR systolic OR diastolic OR SBP OR DBP OR HBP OR HDL OR LDL OR TG OR hyperinsulinemia OR insulin OR "waist circumference" OR overweight OR anthropomorphic OR dyslipidemia OR glucose OR lipid OR cholesterol | 9 |
| | allintitle: Ecuadorian children obesity OR pressure OR systolic OR diastolic OR SBP OR DBP OR HBP OR HDL OR LDL OR TG OR hyperinsulinemia OR insulin OR "waist circumference" OR overweight OR anthropomorphic OR dyslipidemia OR glucose OR lipid OR cholestero | 3 |
| | allintitle: Paraguay children obesity OR pressure OR systolic OR diastolic OR SBP OR DBP OR HBP OR HDL OR LDL OR TG OR hyperinsulinemia OR insulin OR "waist circumference" OR overweight OR anthropomorphic OR dyslipidemia OR glucose OR lipid OR cholesterol | 0 |
| | allintitle: Paraguayan children obesity OR pressure OR systolic OR diastolic OR SBP OR DBP OR HBP OR HDL OR LDL OR TG OR hyperinsulinemia OR insulin OR "waist circumference" OR overweight OR anthropomorphic OR dyslipidemia OR glucose OR lipid OR cholesterol | 0 |
| | allintitle: Peru children obesity OR pressure OR systolic OR diastolic OR SBP OR DBP OR HBP OR HDL OR LDL OR TG OR hyperinsulinemia OR insulin OR "waist circumference" OR overweight OR anthropomorphic OR dyslipidemia OR glucose OR lipid OR cholesterol | 15 |
| | allintitle: Peruvian children obesity OR pressure OR systolic OR diastolic OR SBP OR DBP OR HBP OR HDL OR LDL OR TG OR hyperinsulinemia OR insulin OR "waist circumference" OR overweight OR anthropomorphic OR dyslipidemia OR glucose OR lipid OR cholesterol | 9 |
| | allintitle: Suriname children obesity OR pressure OR systolic OR diastolic OR SBP OR DBP OR HBP OR HDL OR LDL OR TG OR hyperinsulinemia OR insulin OR "waist circumference" OR overweight OR anthropomorphic OR dyslipidemia OR glucose OR lipid OR cholesterol | 2 |
| | allintitle: Surinamese children obesity OR pressure OR systolic OR diastolic OR SBP OR DBP OR HBP OR HDL OR LDL OR TG OR hyperinsulinemia OR insulin OR "waist circumference" OR overweight OR anthropomorphic OR dyslipidemia OR glucose OR lipid OR cholesterol | 4 |
| | allintitle: Uruguay children obesity OR pressure OR systolic OR diastolic OR SBP OR DBP OR HBP OR HDL OR LDL OR TG OR hyperinsulinemia OR insulin OR "waist circumference" OR overweight OR anthropomorphic OR dyslipidemia OR glucose OR lipid OR cholesterol | 1 |
| | allintitle: Uruguayan children obesity OR pressure OR systolic OR diastolic OR SBP OR DBP OR HBP OR HDL OR LDL OR TG OR hyperinsulinemia OR insulin OR "waist circumference" OR overweight OR anthropomorphic OR dyslipidemia OR glucose OR lipid OR cholesterol | 2 |
| | allintitle: Venezuela children obesity OR pressure OR systolic OR diastolic OR SBP OR DBP OR HBP OR HDL OR LDL OR TG OR hyperinsulinemia OR insulin OR "waist circumference" OR overweight OR anthropomorphic OR dyslipidemia OR glucose OR lipid OR cholesterol | 11 |
| | allintitle: Venezuelan children obesity OR pressure OR systolic OR diastolic OR SBP OR DBP OR HBP OR HDL OR LDL OR TG OR hyperinsulinemia OR insulin OR "waist circumference" OR overweight OR anthropomorphic OR dyslipidemia OR glucose OR lipid OR cholesterol | 2 |
| | allintitle: Falkland island children obesity OR pressure OR systolic OR diastolic OR SBP OR DBP OR HBP OR HDL OR LDL OR TG OR hyperinsulinemia OR insulin OR "waist circumference" OR overweight OR anthropomorphic OR dyslipidemia OR glucose OR lipid OR cholesterol | 0 |
| | allintitle: French Guiana children obesity OR pressure OR systolic OR diastolic OR SBP OR DBP OR HBP OR HDL OR LDL OR TG OR hyperinsulinemia OR insulin OR "waist circumference" OR overweight OR anthropomorphic OR dyslipidemia OR glucose OR lipid OR cholesterol | 0 |

*Due to a Google Scholar server error, only 980 of the 1060 results were imported.

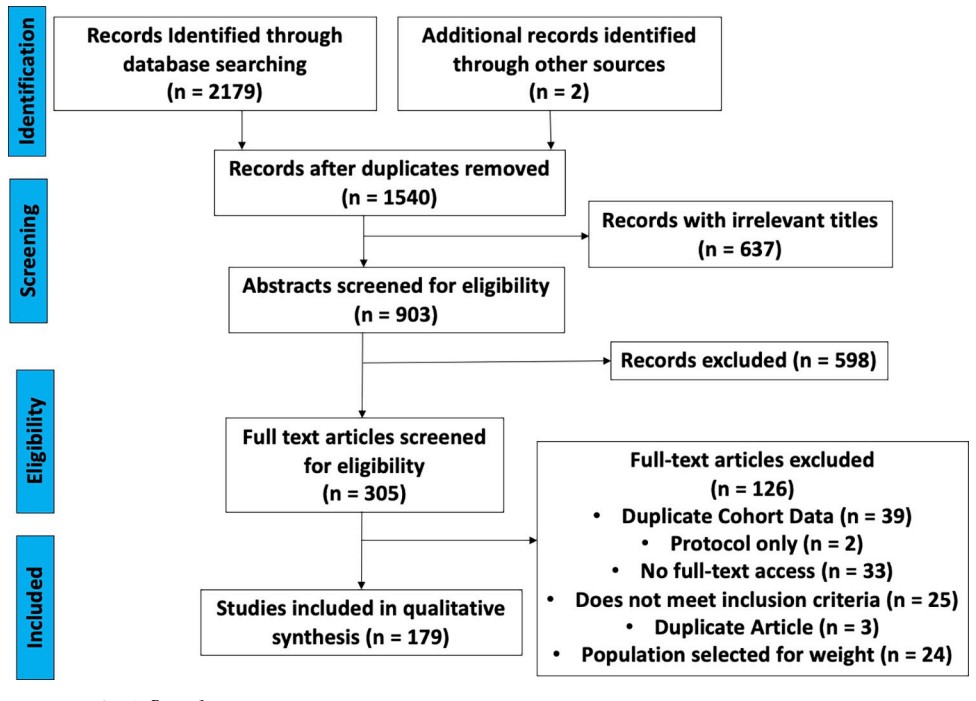

**Fig 1. PRISMA flow chart.**

After comparison of extracted data, disputes were settled through an arbitrator (SLA). For studies that included an intervention, only baseline data was recorded so that interventions could not skew initial population prevalence of CMRF.

## Outcomes

Primary outcomes were prevalence of CMRF and study quality. CMRF definitions used by each study were also recorded. Due to the high amount of heterogeneity in the definitions used, we completed separate analyses with the most common definition for each CMRF to reduce heterogeneity as much as possible. Secondary outcomes were demographic data and study characteristics including age; study setting: rural, mixed, or urban; inclusion of indigenous population; year(s) of data collection; gender distribution: male or female; study location; study design; and study inclusion and exclusion criteria.

## Study quality assessment

Study quality was assessed using a modified Effective Public Healthcare Panacea Project Quality Assessment Tool for Quantitative Studies [11]. Studies were graded on selection bias and data collection methods, then graded as either "Strong," "Moderate," or "Weak" based on their pooled scores. Quality ratings for included titles are presented in Table 2.

## Statistical analysis

We performed descriptive statistics with Microsoft® Excel for Mac Version 16.54 and descriptive and analytical statistics with SPSS 28. Due to the risk of confounding variables and the influence of study heterogeneity, we performed a summative sub-analysis of median prevalence rates of CMRF using the same definition. We used nonparametric, univariate, and

**Table 2. Included studies assessing cardiometabolic risk factors in South American children.**

| Author (et al), year | Years of data collection | Country | City/Region | Study Setting | Risk Factor(s) Studied | Selection Bias Rating | Data Collection Methods Rating | Overall Rating |
|---|---|---|---|---|---|---|---|---|
| Abril, 2013 [12] | 2010–2011 | Ecuador | Cuenca | Urban | Obesity, WC | Strong | Strong | Strong |
| Aglony, 2009 [13] | 2005–2006 | Chile | Santiago | Urban | Obesity, HBP | Moderate | Strong | Strong |
| Aguilar Salinas, 2016 [14] | 2015 | Ecuador | Quito and Cariamanga | Mixed | Obesity, WC | Moderate | Strong | Strong |
| Albuquerque, 2018 [15] | 2016 | Brazil | Viçosa | Urban | Obesity, WC, GI, Dyslipidemia, HBP | Moderate | Moderate | Moderate |
| Alexius, 2012 [16] | Not Reported | Brazil | Medianeira | Mixed | Obesity | Strong | Strong | Strong |
| Alves, 2009 [17] | 2006 | Brazil | Recife | Urban | Obesity | Strong | Moderate | Strong |
| Andaki, 2018 [18] | Not Reported | Brazil | Uberaba | Mixed | Obesity, GI, Dyslipidemia, HBP | Moderate | Strong | Strong |
| Andrade, 2014 [19] | 2008–2009 | Ecuador | Cuenca and Nabon | Mixed | Dyslipidemia | Strong | Strong | Strong |
| Andrade de Medieros Moreira, 2020 [20] | 2018 | Brazil | Palmas | Urban | Obesity | Moderate | Moderate | Moderate |
| Araujo, 2017 [21] | 2012–2013 | Brazil | Piracicaba | Urban | Obesity | Moderate | Weak | Moderate |
| Araujo, 2016 [22] | Not Reported | Brazil | São Caetano do Sul | Urban | Obesity | Moderate | Weak | Moderate |
| Arias Téllez, 2018 [23] | 2008–2011 | Chile | Santiago | Urban | Obesity, WC | Weak | Moderate | Moderate |
| Aristizábal, 2019 [24] | 2015 | Colombia | Medellín | Mixed | Obesity, WC | Weak | Strong | Moderate |
| Assis, 2006 [25] | 2002 | Brazil | Florianópolis | Urban | Obesity | Moderate | Strong | Strong |
| Assunçáo, 2015 [26] | 2005 | Brazil | multiple states | Mixed | Obesity | Moderate | Moderate | Moderate |
| Barbalho, 2017 [27] | Not Reported | Brazil | Lins | Mixed | Obesity, WC, GI, Dyslipidemia | Weak | Strong | Strong |
| Barja, 2011 [28] | 2009–2010 | Chile | Puente Alto | Urban | Obesity, WC, GI | Moderate | Strong | Strong |
| Barros Costa, 2003 [29] | 2000 | Brazil | Juiz de Fora | Mixed | Obesity | Moderate | Moderate | Moderate |
| Bauce, 2018 [30] | 2010–2011 | Venezuela | Caracas | Urban | Obesity | Moderate | Moderate | Moderate |
| Bénéfice*, 2007 [31] | 2004–2005 | Bolivia | Beni River/ Amazon | Rural | Obesity | Moderate | Strong | Strong |
| Benini, 2017 [32] | 2011–2012 | Brazil | Garibaldi | Urban | Obesity, WC, GI, Dyslipidemia | Moderate | Strong | Strong |
| Berghtein, 2014 [33] | 2012 | Argentina | Río Grande, Tierra del Fuego | Mixed | Obesity | Strong | Moderate | Strong |
| Berria, 2013 [34] | 2006 | Brazil | Cascavel | Urban | Obesity, WC | Strong | Moderate | Strong |
| Brinkman, 2015 [35] | 2013 | Chile | Punta Arenas | Urban | Obesity | Strong | Moderate | Strong |
| Bozzini, 2019 [36] | 2014–2017 | Brazil | São Paulo | Urban | Obesity, Dyslipidemia | Moderate | Moderate | Moderate |
| Buitrago-Lopez, 2015 [37] | 2006–2007 | Colombia | Bucaramanga | Urban | Obesity | Moderate | Strong | Strong |
| Burgos, 2019 [38] | 2011–2012 | Brazil | Santa Cruz do Sul | Mixed | Obesity, WC, Dyslipidemia, HBP | Strong | Weak | Moderate |
| Burrows, 2015 [39] | Not Reported | Chile | Santiago | Urban | Obesity, WC, GI, Dyslipidemia, HBP | Weak | Strong | Moderate |
| Bustamante, 2013 [40] | 2009–2010 | Peru | Central region | Mixed | Obesity, WC | Strong | Moderate | Strong |
| Caamaño Navarrete, 2015 [41] | Not Reported | Chile | Temuco | Urban | Obesity | Moderate | Moderate | Strong |
| Caixeta, 2020 [42] | 2016–2017 | Brazil | Brasilia | Urban | Obesity, WC | Strong | Moderate | Strong |
| Calvo, 2004 [43] | 2002 | Brazil | Florianopolis | Urban | Obesity | Strong | Moderate | Strong |
| Campos, 2006 [44] | 2003 | Brazil | Fortaleza | Urban | Obesity | Moderate | Moderate | Moderate |
| Campoverde Ríos, 2016 [45] | Not Reported | Ecuador | Quito | Urban | Obesity, WC, GI | Moderate | Moderate | Moderate |

(*Continued*)

**Table 2.** (Continued)

| Author (et al), year | Years of data collection | Country | City/Region | Study Setting | Risk Factor(s) Studied | Selection Bias Rating | Data Collection Methods Rating | Overall Rating |
|---|---|---|---|---|---|---|---|---|
| Carolina Avalos, 2012 [46] | 2006–2007 | Chile | Santiago | Urban | Obesity | Moderate | Strong | Strong |
| Carrillo-Larco, 2016 [47] | 2006–2007 | Peru | | Mixed | Obesity | Moderate | Moderate | Moderate |
| Casagrande, 2017 [48] | 2015 | Brazil | Marília | Mixed | Obesity, WC | Moderate | Strong | Strong |
| Casapulla, 2017 [49] | 2015 | Ecuador | Pomasqui and Cariamanga | Urban | Obesity, GI | Moderate | Strong | Strong |
| Castro Burbano, 2016 [50] | Not Reported | Ecuador | | Mixed | Obesity | Moderate | Moderate | Moderate |
| Cediel, 2016 [51] | 2002- ongoing | Chile | Santiago | Urban | Obesity, WC, GI | Moderate | Strong | Strong |
| Cesani, 2020 [52] | 2015–2017 | Artentina | La Plata | Mixed | Obesity, HBP | Strong | Strong | Strong |
| Cobo, 2015 [53] | 2011 | Chile | Quillota | Mixed | Obesity, HBP | Moderate | Moderate | Moderate |
| Cohen, 2016 [54] | 2012 | Argentina | Pilar Buenos Aires | Mixed | Obesity, HBP | Moderate | Strong | Strong |
| Cohen, 2014 [55] | 2011–2012 | Colombia | Bucaramanga | Urban | Obesity | Moderate | Moderate | Moderate |
| Collazo, 2018 [56] | 2016–2017 | Ecuador | Cuenca | Mixed | Obesity | Moderate | Moderate | Moderate |
| Corso, 2004 [57] | 2002 | Brazil | Floranópolis | Mixed | Obesity | Moderate | Moderate | Moderate |
| Corvalán, 2013 [58] | 2002- ongoing | Chile | Santiago | Urban | Obesity, WC | Strong | Strong | Strong |
| Corvalan, 2010 [59] | 2006 | Chile | Santiago | Urban | Obesity, WC, GI, Dyslipidemia | Moderate | Strong | Strong |
| Costa, 2006 [60] | 2002 | Brazil | Santos | Urban | Obesity | Strong | Moderate | Strong |
| Crovetto, 2010 [61] | 2008 | Chile | Valparíso | Urban | Obesity | Strong | Weak | Moderate |
| Cruz, 2017 [62] | 2008 | Brazil | Pelotas | Urban | Obesity | Moderate | Weak | Moderate |
| da Silva, 2013 [63] | 2005–2006 | Brazil | Alagotas State | Mixed | Obesity | Moderate | Strong | Strong |
| de Carvahlo Cremm, 2011 [64] | Not Reported | Brazil | Santos | Urban | Obesity | Moderate | Moderate | Moderate |
| de Melo, 2016 [65] | 2010–2012 | Brazil | Natal | Urban | Obesity, WC, GI, Dyslipidemia, HBP | Moderate | Moderate | Moderate |
| Delgado-Floody, 2019 [66] | Not Reported | Chile | Araucania Region | Mixed | Obesity, HBP | Moderate | Moderate | Moderate |
| Delgado-Floody, 2017 [67] | Not Reported | Chile | | Mixed | Obesity, WC | Moderate | Strong | Strong |
| De Santis Filgueiras, 2018 [68] | 2015 | Brazil | Viçosa, Minas Gerais | Mixed | Obesity, WC | Strong | Strong | Strong |
| De Santis Filgueiras, 2018 [69] | 2015 | Brazil | Viçosa, Minas Gerais | Mixed | Dyslipidemia | Strong | Strong | Strong |
| Devia Solia, 2018 [70] | 2017–2018 | Ecuador | Azogues | Urban | Obesity, WC, Dyslipidemia, HBP | Weak | Strong | Moderate |
| Diaz, 2010 [71] | Not Reported | Argentina | Maria Ignacia Vela | Rural | Obesity, HBP | Moderate | Moderate | Moderate |
| Duncan, 2011 [72] | Not Reported | Brazil | São Paulo state | Mixed | Obesity | Strong | Moderate | Strong |
| Escalona-Villasmil, 2016 [73] | 2013 | Venezuela | Marcaibo | Urban | Obesity | Weak | Strong | Moderate |
| Ferrari, 2019 [74] | Not Reported | Brazil | São Caetano do Sul | Urban | Obesity | Moderate | Strong | Strong |
| Ferreira, 2009 [75] | Not Reported | Brazil | Taguatinga, Brasília | Mixed | Obesity | Moderate | Strong | Strong |
| Figueroa Sobero, 2016 [76] | Not Reported | Argentina | Riós, Buenos Aires, CABC, Salta, Córdoba, Tierra del Fuego, Corrientes | Urban | Obesity, WC, GI, Dyslipidemia, HBP | Moderate | Strong | Strong |
| Florencio de Souza, 2011 [77] | 2004 | Brazil | Rio Branco, Acre State | Mixed | Obesity | Strong | Moderate | Strong |
| Fornasini, 2016 [78] | Not Reported | Ecuador | Quito | Urban | Obesity | Moderate | Weak | Moderate |

(*Continued*)

**Table 2.** (Continued)

| Author (et al), year | Years of data collection | Country | City/Region | Study Setting | Risk Factor(s) Studied | Selection Bias Rating | Data Collection Methods Rating | Overall Rating |
|---|---|---|---|---|---|---|---|---|
| Gamboa-Delgado, 2017 [79] | 2006–2007 | Colombia | Bucaramanga | Urban | Obesity, GI, Dyslipidemia, HBP | Moderate | Strong | Strong |
| Game, 2015 [80] | Not Reported | Chile | Boyeco | Urban | Obesity | Moderate | Weak | Moderate |
| Garcia-Hermoso, 2019 [81] | 2016–2017 | Chile | Santiago | Urban | Obesity, HBP | Moderate | Moderate | Moderate |
| Gaya, 2017 [82] | 2008–2009 and 2013–2014 | Brazil | 16 states | Mixed | Obesity | Moderate | Strong | Strong |
| Gilbert-Diamond, 2012 [83] | 2006 | Colombia | Bogotá | Urban | Obesity | Strong | Strong | Strong |
| Giuliano, 2011 [84] | 2002–2004 | Brazil | Florianópolis | Urban | Dyslipidemia | Moderate | Moderate | Strong |
| Gómes, 2019 [85] | 2008–2015 | Brazil | Campinas | Mixed | Dyslipidemia | Strong | Strong | Strong |
| Gomes, 2017 [86] | 2004–2005 | Brazil | Feria de Santana | Mixed | Obesity | Moderate | Strong | Strong |
| Gomez, 2007 [87] | 2005 | Colombia | | Mixed | Obesity | Moderate | Moderate | Moderate |
| Granville-Garcia, 2008 [88] | Not Reported | Brazil | Recife | Urban | Obesity | Moderate | Moderate | Moderate |
| Guedes, 2013 [89] | 2011 | Brazil | Monte Claros | Mixed | Obesity | Strong | Strong | Strong |
| Guevara, 2018 [90] | 2015–2016 | Ecuador | Riobamba | Urban | Obesity | Moderate | Weak | Moderate |
| Gutierez-Gomez, 2009 [91] | 2006 | Chile | | Mixed | Obesity, WC | Strong | Strong | Strong |
| Hércules, 2020 [92] | 2016–2017 | Brazil | Curibata | Mixed | Obesity | Moderate | Moderate | Moderate |
| Herrán, 2017 [93] | 2009–2011 | Colombia | | Mixed | Obesity | Moderate | Moderate | Moderate |
| Herrera Sevilla, 2015 [94] | 2015 | Ecuador | Quito | Urban | Obesity | Weak | Weak | Weak |
| Hirshler*, 2014 [95] | 2011–2013 | Argentina | San Antonio de los Cobres | Rural | Obesity | Moderate | Moderate | Moderate |
| Hirschler*, 2012 [96] | 2007–2008 | Argentina | San Antonio de los Cobres and Buenos Aires | Mixed | Obesity, WC, GI, Dyslipidemia, HBP | Moderate | Strong | Strong |
| Hirschler, 2009 [97] | 2007 | Argentina | Buenos Aires | Urban | Obesity | Moderate | Moderate | Moderate |
| Hirschler, 2010 [98] | 2007 | Argentina | Buenos Aires | Urban | Obesity, WC, GI, Dyslipidemia, HBP | Strong | Strong | Strong |
| Hirschler*, 2016 [99] | 2011–2014 | Argentina | San Antonio de los Cobres | Rural | Obesity, Dyslipidemia | Strong | Moderate | Strong |
| Hirschler, 2006 [100] | 2004 | Argentina | Buenos Aires | Urban | Obesity | Moderate | Moderate | Moderate |
| Hirschler*, 2015 [101] | 2012–2013 | Argentina | San Antonio de los Cobres and Buenos Aires | Mixed | Obesity, Dyslipidemia | Strong | Moderate | Strong |
| Hirschler*, 2013 [102] | Not Reported | Argentina | San Antonio de los Cobres | Rural | Obesity | Strong | Strong | Strong |
| Honório, 2014 [103] | 2011–2012 | Brazil | Goiânia | Urban | Obesity | Moderate | Strong | Strong |
| Houck, 2017 [104] | Not Reported | Ecuador | Galapagos | Mixed | Obesity | Moderate | Weak | Moderate |
| Houck, 2013 [105] | 2009 | Ecuador | Northern Ecuador Amazon | Rural | Obesity | Moderate | Strong | Moderate |
| Iguarán Kohen*, 2015 [106] | 2015 | Colombia | Siapana | Rural | Obesity, WC | Moderate | Strong | Strong |
| Inciarte, 2013 [107] | Not Reported | Venezuela | Maracaibo | Urban | Obesity | Moderate | Moderate | Moderate |
| Jones, 2018 [108] | 2015 | Bolivia | El Alto and Montero | Mixed | Obesity | Moderate | Strong | Strong |
| Kain, 2014 [109] | 2011–2012 | Chile | Nuñoa | Mixed | Obesity | Strong | Weak | Moderate |
| Kain, 2007 [110] | 2002–2004 | Chile | Santiago | Mixed | Obesity | Moderate | Weak | Moderate |
| Kain, 2016 [111] | 2006 onward | Chile | | Mixed | Obesity, WC | Moderate | Moderate | Moderate |
| Kain, 2009 [112] | 2006 | Chile | Santiago | Urban | Obesity | Strong | Strong | Strong |
| Kain, 2009 [113] | 2003 | Chile | Casablanca and Quillota | Mixed | Obesity | Moderate | Strong | Strong |
| Kupek, 2014 [114] | 2007 | Brazil | Florianopolis | Mixed | Obesity | Moderate | Moderate | Moderate |

(Continued)

**Table 2.** (Continued)

| Author (et al), year | Years of data collection | Country | City/Region | Study Setting | Risk Factor(s) Studied | Selection Bias Rating | Data Collection Methods Rating | Overall Rating |
|---|---|---|---|---|---|---|---|---|
| Leal, 2012 [115] | 2006 | Brazil | Pernambuco State | Mixed | Obesity | Strong | Moderate | Strong |
| Leite, 2017 [116] | 2001–2014 | Brazil | Juiz de Fora | Mixed | Obesity | Moderate | Weak | Moderate |
| Linares Herrera, 2017 [117] | 2015 | Peru | Tarapoto | Urban | Obesity, WC | Moderate | Strong | Strong |
| Lizana, 2016 [118] | 2013–2014 | Chile | Quilota and Valparaíso | Mixed | Obesity, WC | Moderate | Weak | Moderate |
| Loaiza, 2012 [119] | Not Reported | Chile | | Mixed | Obesity | Moderate | Weak | Moderate |
| Lourenço, 2014 [120] | 2007–2009 | Brazil | Acrelândia | Mixed | Obesity | Moderate | Strong | Strong |
| Matos, 2011 [121] | 2005 | Brazil | Salvador | Urban | Obesity, GI | Moderate | Strong | Strong |
| Morillo Silva, 2017 [122] | Not Reported | Ecuador | Santo Domingo | Urban | Obesity | Weak | Weak | Weak |
| Muñoz, 2016 [123] | Not Reported | Colombia | Medellín | Urban | Obesity, WC, GI, Dyslipidemia, HBP | Moderate | Strong | Strong |
| Musso, 2011 [124] | 2008 | Argentina | Buenos Aires | Urban | Obesity, WC, GI, Dyslipidemia, HBP | Moderate | Strong | Strong |
| Naves, 2006 [125] | 2005 | Brazil | Brasilia | Urban | Dyslipidemia | Moderate | Strong | Strong |
| Neyla de Lima Albaquerque, 2016 [126] | 2013 | Brazil | Recife | Urban | Obesity, WC, Dyslipidemia | Weak | Strong | Moderate |
| Ninatana-Ortiz, 2016 [127] | 2014 | Peru | Cajamarca | Mixed | Obesity, WC, GI, Dyslipidemia, HBP | Moderate | Strong | Strong |
| Novaes, 2011 [128] | 2005–2006 | Brazil | Visçosa | Mixed | Obesity | Moderate | Moderate | Moderate |
| Obregón, 2018 [129] | 2014–2015 | Chile | Concepción | Mixed | Obesity | Strong | Moderate | Moderate |
| Olaya-Contreras, 2015 [130] | 2013 | Colombia | Medellin | Urban | Obesity | Moderate | Moderate | Strong |
| Oliveira Pani, 2015 [131] | Not Reported | Brazil | Espiritu Santo province | Urban | Obesity, Dyslipidemia | Weak | Weak | Weak |
| Oliveira, 2007 [132] | Not Reported | Brazil | Feria de Santana | Urban | Obesity | Moderate | Moderate | Moderate |
| Orden, 2018 [133] | 2015–2016 | Argentina | Santa Rosa | Urban | Obesity | Moderate | Strong | Strong |
| Padula, 2012 [134] | 2003–2005 | Argentina | La Plata, Buenos Aires | Urban | Obesity | Moderate | Strong | Strong |
| Pajuelo-Ramírez, 2013 [135] | 2009–2010 | Peru | | Mixed | Obesity | Moderate | Moderate | Moderate |
| Pajuelo-Ramírez, 2011 [136] | 2007–2010 | Peru | | Mixed | Obesity | Moderate | Moderate | Moderate |
| Passos, 2015 [137] | 2012 | Brazil | Pelotas | Mixed | Obesity | Moderate | Strong | Strong |
| Peregalli, 2013 [138] | Not Reported | Uruguay | | Mixed | Obesity | Moderate | Weak | Moderate |
| Pereira, 2009 [139] | 2001 | Brazil | Itapetininga | Mixed | Obesity, Dyslipidemia, HBP | Moderate | Moderate | Moderate |
| Pereira, 2013 [140] | 2004–2006 | Brazil | Jundiai | Urban | Obesity, Dyslipidemia | Moderate | Weak | Moderate |
| Pereyra, 2021 [141] | 2013–2014 and 2015–2016 | Uruguay | | Mixed | Obesity | Moderate | Weak | Moderate |
| Pincón, 2011 [142] | 2010–2011 | Ecuador | Cuenca | Rural | Obesity | Weak | Moderate | Moderate |
| Poveda, 2007 [143] | Not Reported | Colombia | | Mixed | Obesity | Moderate | Moderate | Moderate |
| Quadros, 2016 [144] | 2011–2012 | Brazil | Amargosa | Mixed | Obesity, WC, GI, Dyslipidemia, HBP | Moderate | Moderate | Moderate |
| Ramirez-Velez, 2017 [145] | 2013–2016 | Colombia | Bogota | Urban | Obesity, WC, GI, Dyslipidemia, HBP | Moderate | Strong | Strong |
| Ramirez-Velez, 2016 [146] | 2013–2016 | Colombia | Bogotá | Urban | Obesity | Moderate | Strong | Strong |

(*Continued*)

**Table 2.** (Continued)

| Author (et al), year | Years of data collection | Country | City/Region | Study Setting | Risk Factor(s) Studied | Selection Bias Rating | Data Collection Methods Rating | Overall Rating |
|---|---|---|---|---|---|---|---|---|
| Ramos-Padilla, 2015 [147] | 2013 | Ecuador | Riobamba | Urban | Obesity | Moderate | 0 | Moderate |
| Real Delor, 2017 [148] | 2020 | Paraguay | Asunción | Urban | Obesity, HBP | Moderate | Moderate | Moderate |
| Ribas, 2012 [149] | 2005 | Brazil | Belém | Urban | Obesity, WC, Dyslipidemia | Moderate | Weak | Moderate |
| Ribeiro, 2017 [150] | Not Reported | Brazil | São Luís | Urban | Obesity | Moderate | Strong | Strong |
| Ricardo, 2009 [151] | 2007–2008 | Brazil | Santa Catarina state | Mixed | Obesity | Strong | Strong | Strong |
| Rinaldi, 2010 [152] | 2007 | Brazil | Botucatu | Urban | Obesity | Moderate | Strong | Strong |
| Rincón, 2015 [153] | 2010–2011 | Vanezuela | Merida | Urban | Obesity, GI, Dyslipidemia, HBP | Strong | Strong | Strong |
| Rizzo, 2013 [154] | 2009–2011 | Brazil | Botucatu | Urban | Obesity, WC, GI, Dyslipidemia, HBP | Moderate | Moderate | Moderate |
| Rodrigues-Bezerra, 2016 [155] | 2014–2015 | Colombia | Bogotá | Urban | Obesity | Strong | Strong | Strong |
| Romagna, 2010 [156] | 2007–2008 | Brazil | Canoes | Mixed | Obesity | Moderate | Strong | Strong |
| Romero-Sandoval, 2012 [157] | 2010–2011 | Ecuador | Quito | Urban | Obesity | Strong | Moderate | Strong |
| Rosini, 2013 [158] | 2009 | Brazil | Guabiruba | Mixed | Obesity, WC | Weak | Strong | Moderate |
| Rossi, 2018 [159] | Not Reported | Ecuador | Galápagos | Urban | Obesity | Moderate | Moderate | Moderate |
| Ruiz, 2014 [160] | 2012–2013 | Venezuela | Valencia | Urban | Obesity, WC, GI | Weak | Moderate | Moderate |
| Salazar-Guitérrez, 2020 [161] | Not Reported | Chile | Chillán | Mixed | Obesity | Moderate | Moderate | Moderate |
| Salceda, 2013 [162] | 2004 | Argentina | Catamarca | Mixed | Obesity | Moderate | Moderate | Moderate |
| Saldiva, 2004 [163] | 2001 | Brazil | Bady Bassit, Bofete, Jaborandi, Morungaba, Riversul | Mixed | Obesity | Strong | Strong | Strong |
| Saldiva, 2007 [164] | 2001 | Brazil | Bady Bassit, Bofete, Jaborandi, Morungaba, Riversul | Mixed | Obesity | Moderate | Strong | Strong |
| Santos, 2019 [165] | 2009–2010 | Peru | Barranco, La Merced, San Ranom, Junín | Mixed | Obesity | Moderate | Moderate | Moderate |
| Sapunar, 2018 [166] | 2015–2016 | Chile | Carahue | Urban | Obesity, WC, GI, Dyslipidemia, HBP | Moderate | Weak | Moderate |
| Sehn, 2016 [167] | Not Reported | Brazil | Santa Cruz do Sul | Urban | Obesity | Moderate | Weak | Moderate |
| Sentalin, 2019 [168] | 2015 | Brazil | Vinhedo | Mixed | Obesity, WC | Moderate | Moderate | Moderate |
| Serrano, 2019 [169] | 2006–2017 | Colombia | Bucaramanga | Urban | Obesity, WC, GI, Dyslipidemia, HBP | Weak | Strong | Moderate |
| Silva, 2018 [170] | 2012–2013 | Brazil | Uberaba | Mixed | Obesity | Moderate | Moderate | Moderate |
| Silva, 2011 [171] | 2008–2010 | Brazil | Sergipe | Mixed | Obesity | Moderate | Weak | Moderate |
| Silva, 2009 [172] | 2005 | Brazil | João Pessoa | Mixed | HBP | Strong | Strong | Strong |
| Silveira, 2014 [173] | 2006–2007 | Brazil | | Mixed | Obesity | Moderate | Moderate | Moderate |
| Solano, 2003 [174] | 2002 | Venezuela | Valencia | Urban | Obesity, Dyslipidemia | Moderate | Weak | Moderate |
| Souza, 2021 [175] | 2016 | Brazil | Monte Negro | Mixed | Obesity, Dyslipidemia, HBP | Moderate | Strong | Strong |
| Suarez-Lopez, 2019 [176] | 2008 | Ecuador | Pedro Moncayo County | Rural | HBP | Moderate | Strong | Strong |
| Suarez-Ortegón, 2016 [177] | Not Reported | Colombia | Cali | Urban | Obesity, WC, GI, Dyslipidemia, HBP | Moderate | Strong | Strong |

(*Continued*)

**Table 2.** (Continued)

| Author (et al), year | Years of data collection | Country | City/Region | Study Setting | Risk Factor(s) Studied | Selection Bias Rating | Data Collection Methods Rating | Overall Rating |
|---|---|---|---|---|---|---|---|---|
| Tande da Silva, 2011 [178] | 2007–2009 | Brazil | São Paulo | Urban | Obesity | Weak | Strong | Moderate |
| Tardivo, 2013 [179] | Not Reported | Brazil | Sao Paulo | Mixed | Obesity | Moderate | Weak | Moderate |
| Tarqui-Mamani, 2018 [180] | 2013–2014 | Peru | | Mixed | Obesity | Moderate | Moderate | Moderate |
| Todendi, 2019 [181] | 2014–2015 | Brazil | Santa Cruz do sul | Mixed | Obesity, WC | Moderate | Moderate | Moderate |
| Torres-Roman, 2018 [182] | 2010–2015 | Peru | | Mixed | Obesity | Moderate | Moderate | Moderate |
| Torres, 2015 [183] | 2010–2011 | Brazil | Maruípe, Continebtal | Mixed | Obesity | Moderate | Weak | Moderate |
| Toso, 2003 [184] | Not Reported | Brazil | Jaragua do sul | Mixed | Obesity | Moderate | Weak | Moderate |
| Ulloa, 2010 [185] | 2006 | Chile | Concepción, Coronel, Hualpén | Urban | Obesity | Strong | Weak | Moderate |
| Vanzelli, 2008 [186] | 2005 | Brazil | Jundiaí | Urban | Obesity | Moderate | Strong | Strong |
| Verona, 2013 [187] | Not Reported | Argentina | Balcarce | Rural | Obesity, WC, GI, Dyslipidemia, HBP | Moderate | Strong | Strong |
| Villalba-Condori, 2019 [188] | Not Reported | Peru | | Mixed | Obesity | Weak | Weak | Weak |
| Zaccarelli-Marino, 2020 [189] | 2011–2012 | Brazil | Santo André | Urban | Obesity | Moderate | Moderate | Moderate |
| Zandoná, 2017 [190] | 2001–2005, 2008 | Brazil | Leopoldo and Porto Alegre | Mixed | Obesity | Moderate | Moderate | Moderate |

*indicates study included indigenous population; WC = waist circumference; GI = glucose intolerance; HBP = high blood pressure.

multivariate testing. For variables with two groups, we utilized the Independent-Samples Mann-Whitney U Test, and for variables with more than two groups we used the Independent-Samples Kruskal-Wallis Test.

## Results

### Study characteristics and population demographics

Of 2,181 studies screened, we included 179 studies. These were comprised of 2,975,261 unique subjects representing ten countries (Argentina, Bolivia, Brazil, Chile, Colombia, Ecuador, Paraguay, Peru, Uruguay, and Venezuela). Our search did not find relevant results for the following countries and territories in South America: Guyana, Suriname, French Guiana, and the Falkland Islands. Peru had the largest population with 2,380,473 subjects, followed by Brazil with 276,648. Paraguay had the least with 132 subjects. The range of female subjects was 0–100% with median of 51.0%. Eight studies (4.5%) included indigenous populations. Study settings were 6.2% rural, 48.0% urban, and 45.8% mixed settings.

### Cardiometabolic risk factor definitions

The definitions for each risk factor are presented in Table 3. Included studies used 15 different definitions for obesity, 8 definitions for elevated waist circumference, 4 definitions for impaired fasting glucose, 11 definitions for elevated triglycerides, 9 definitions for low HDL cholesterol, and 8 definitions for HBP.

**Table 3. All definitions used for cardiometabolic risk factors in studies assessing South American Children.**

| Risk Factor | Definitions Used (n = number of studies used definition) |
|---|---|
| Obesity (159) | BMI, WH, WL ≥95% for age and gender (53, 33.3%);<br>BMIZ ≥ +2 above mean (45, 28.3%);<br>Country- or Study-specific percentiles (12, 7.5%);<br>BMI, WH, WL ≥97% for age and gender (11, 6.9%);<br>International Obesity Task Force Percentiles (10, 6.3%);<br>Cole et al. BMI percentiles (8, 5.0%);<br>BMIZ, WLZ, WHZ ≥+3 (5, 3.1%);<br>BMI ≥30 (1, 0.6%)<br>Unknown (8, 5.0%) |
| Elevated Waist Circumference (n = 43) | WC ≥90% for age, height and gender (29, 67.4%);<br>WC ≥75% for age, height and gender (5, 11.6%);<br>Country- or Study-Specific Percentiles (4, 9.3%);<br>WC ≥80% for height and age (2, 4.6%);<br>Cutoffs proposed by Taylor et al. (2, 4.6%);<br>WC ≥80 and 90 cm in females and males respectively (2, 4.6%);<br>Cutoffs defined from World Health Organization 1988 (1, 2.3%);<br>Ratio of WC:height ≥0.55 (1, 2.3%) |
| Glucose Intolerance (n = 27) | Fasting glucose ≥100 mg/dL (21, 77.8%);<br>Fasting glucose ≥110 mg/dL (4, 14.8%);<br>Fasting glucose ≥126 mg/dL (1, 3.7%);<br>Fasting glucose ≥5.55 mmol/L (1, 3.7%) |
| Elevated Triglycerides (n = 30) | TG ≥150 mg/dL (10, 33.3%);<br>TG ≥100 mg/dL in children age 2–9 yo (10, 33.3%);<br>TG ≥ 130 mg/dL in children ≥10 yo (7, 23.3%);<br>TG ≥110 mg/dL (6, 20.0%);<br>TG ≥1.96 mmol/L (1, 3.3%);<br>TG ≥1.5 mmol/L (1, 3.3%);<br>TG ≥75 mg/dL (1, 3.3%);<br>TG ≥95% for age and gender (1, 3.3%);<br>TG ≥90% for age and gender (1, 3.3%);<br>TG ≥75≤99 mg/dL for ages 0–9 and ≥90≤125 mg/dL for ages 15–19 (1, 3.3%);<br>TG >100 mg/dL for age 2–15 and ≥125 for age 15–19 (1, 3.3%);<br>unknown (1, 3.3%) |
| Low HDL Cholesterol (n = 38) | HDL ≤40 mg/dL (18, 47.4%);<br>HDL ≤45 mg/dL (9, 23.7%);<br>HDL ≤35 mg/dL (4, 10.5%); HDL ≤50 mg/dL (1, 2.6%);<br>HDL ≤50 mg/dL for females and HDL ≤40 mg/dL for males (1, 2.6%);<br>HDL ≤1.2 mmol/L (1, 2.6%);<br>HDL ≤1.09 mmol/L (1, 2.6%);<br>HDL ≤ 0.9 mmol/L (1, 2.6%);<br>HDL <10% for age and gender (1, 2.6%);<br>HDL ≤38 mg/dL for females and ≤36 mg/dL for males (1, 2.6%) |
| High Blood Pressure (n = 33) | SBP or DBP ≤90% for age, height and gender (22, 66.7%);<br>SBP ≥130 mmHg (4, 12.1%);<br>DBP ≥85 mmHg (4, 12.1%);<br>SBP or DBP ≥95% for age, height and gender (4, 12.1%);<br>American Pediatric Society guidelines (1, 3.0%);<br>Argentinean national guidelines (1, 3.0%) |

BMI = body mass index; WH = waist to height ratio; WL = weight to length ratio; WC = waist circumference;

TG = triglycerides; HDL = high density lipoprotein; SBP = systolic blood pressure; DBP = diastolic blood pressure.

## Obesity

Results are summarized in Table 4. One hundred and fifty-nine studies (94.4% of included studies) measured obesity; these included 2,787,579 individuals and 15 different definitions. The most commonly used definitions of obesity were BMI, WH (waist-to-height ratio), or WL

**Table 4. Cardiometabolic risk factors in South American children by any definition and by the most common definition, study setting, country, and inclusion of indigenous population.**

| Variables | | Obesity by any definition | | | Obesity by BMI, WH, WL ≥95%* | | |
|---|---|---|---|---|---|---|---|
| | | N (%) | Prevalence %, median (IQR) | p value | N (%) | Prevalence %, median (IQR) | p value |
| Overall | | 2787579 | 12.2 (6.3–17.3) | | 217989 | 13.4 (7.0–20.6) | |
| Setting | Urban | 105643 | 12.6 (6.5–18.0) | 0.08 | 63398 | 13.9 (6.8–20.4) | 0.08 |
| | Mixed | 2672381 | 12.9 (6.5–16.2) | | 147401 | 13.6 (7.2–17.6) | |
| | Rural | 8702 | 5.6 (1.6–12.0) | | 7190 | 4.9 (2.0–9.1) | |
| Country | Argentina | 1400 | 11.1 (3.4–16.3) | <0.001 | 6567 | 11.1 (4.4–14.6) | 0.03 |
| | Bolivia | 556 | 1.6 | | | | |
| | Brazil | 179591 | 10.1 (6.5–15.9) | | 17609 | 12.8 (7.7–18.0) | |
| | Chile | 192333 | 20.6 (15.6–27.8) | | 183393 | 21.4 (13.7–28.5) | |
| | Colombia | 12633 | 6.5 (4.1–9.7) | | 2628 | 7.4 (2.1) | |
| | Ecuador | 13523 | 6.3 (2.6–17.3) | | 7292 | 5.8 (3.6) | |
| | Paraguay | 132 | 2.3 | | | | |
| | Peru | 2368236 | 8.1 (4.2–14.1) | | 500 | 14.4 | |
| | Uruguay | 4048 | 13.2 (11.6) | | | | |
| | Venezuela | 1674 | 7.3 (4.8–12.8) | | | | |
| Indigenous Population | Yes | 1554 | 2.5 (1.6–5.6) | 0.009 | 812 | 3.4 (1.6) | 0.007 |
| | No | 2786025 | 12.7 (6.5–17.5) | | 217177 | 13.9 (7.7–21.1) | |
| | | Obese and/or overweight by any definition | | | Obesity and/or overweight defined as BMI,WH,WL ≥95% | | |
| Variables | | N (%) | Prevalence %, median (IQR) | p value | N (%) | Prevalence %, median (IQR) | p value |
| Overall | | 504853 | 23.9 (16.5–39.3) | | 502726 | 22.9 (18.0–35.6) | |
| Setting | Urban | 170723 | 26.3 (18.3–41.3) | 0.01 | 116546 | 26.3 (19.0–35.1) | 0.003 |
| | Mixed | 324007 | 22.8 (16.6–35.7) | | 130558 | 20.4 (12.0–24.1) | |
| | Rural | 10123 | 16.2 (10.8–23.6) | | 8518 | 10.9 (6.7–16.1) | |
| Country | Argentina | 13739 | 21.1 (10.1–32.6) | <0.001 | 6135 | 18.4 (10.9–29.8) | 0.008 |
| | Bolivia | 2873 | 10.6 (10.5) | | | | |
| | Brazil | 201223 | 24.0 (17.7–36.2) | | 28563 | 22.8 (16.8–34.6) | |
| | Chile | 166303 | 47.5 (41.1–60.8) | | 157609 | 50.2 (35.0–64.2) | |
| | Colombia | 73249 | 18.7 (11.9–23.4) | | 55326 | 18.1 (13.4–22.6) | |
| | Ecuador | 13720 | 19.9 (13.5–26.4) | | 7489 | 24.7 (21.0–81.6) | |
| | Paraguay | 132 | 10.6 | | | | |
| | Peru | 31447 | 21.7 (20.5–31.6) | | 500 | 22 | |
| | Uruguay | 103 | 27.2 | | | | |
| | Venezuela | 2064 | 20.8 (15.2–29.0) | | | | |
| Indigenous Population | Yes | 2975 | 10.8 (7.2–12.9) | <0.001 | 2140 | 10.9 (6.6–12.2) | <0.01 |
| | No | 501878 | 25.2 (17.9–40.1) | | 253482 | 24.7 (19.9–38.2) | |
| | | Elevated Waist Circumference by any definition | | | Elevated Waist Circumference defined as WC ≥90% for age and gender | | |
| Variables | | N (%) | Prevalence %, median (IQR) | p value | N (%) | Prevalence %, median (IQR) | p value |
| Overall | | 38250 | 21.9 (9.4–33.9) | | 22865 | 10.6 (5.4–27.7) | |
| Setting | Urban | 25416 | 24.0 (8.6–38.1) | 0.55 | 17934 | 10.6 (5.2–28.2) | 0.40 |
| | Mixed | 10166 | 30.4 (10.9–41.0) | | 4099 | 5.6 (3.0) | |
| | Rural | 2668 | 16.3 (10.7–23.8) | | 832 | 16.3 (6.4) | |

(*Continued*)

**Table 4.** (Continued)

| Variables | | Obesity by any definition | | | Obesity by BMI, WH, WL ≥95%* | | |
|---|---|---|---|---|---|---|---|
| | | N (%) | Prevalence %, median (IQR) | p value | N (%) | Prevalence %, median (IQR) | p value |
| Country | Argentina | 4211 | 19.2 (5.1–31.2) | 0.04 | 4211 | 19.2 (5.1–31.3) | 0.26 |
| | Brazil | 14410 | 29.6 (10.5–48.0) | | 4773 | 20.1 (10.2–43.6) | |
| | Chile | 6757 | 26.7 (18.7–35.1) | | 1686 | 26.0 (16.2) | |
| | Colombia | 10337 | 11.1 (4.1–28.5) | | 9750 | 5.6 (4.1–22.0) | |
| | Ecuador | 1359 | 7.6 (3.4–23.6) | | 1359 | 7.6 (3.4–23.6) | |
| | Peru | 1086 | 5.4 (5.2) | | 1086 | 5.4 (5.2) | |
| | Venezuela | 90 | 42.2 | | | | |
| Indigenous Population | Yes | 423 | 10.7 (6.4) | 0.28 | | | |
| | No | 37827 | 24.0 (10.0–35.1) | | | | |
| | | **Impaired Fasting Glucose by any definition** | | | **Impaired Fasting Glucose by Glucose ≥100 mg/dL** | | |
| Variables | | N (%) | Prevalence %, median (IQR) | p value | N (%) | Prevalence %, median (IQR) | p value |
| Overall | | 19810 | 3.0 (1.0–6.6) | | 15263 | 2.8 (1.0–6.7) | |
| Setting | Urban | 11941 | 3.2 (1.5–6.2) | 0.92 | 10908 | 2.8 (1.4–6.7) | 0.10 |
| | Mixed | 5076 | 3.7 (0.6–8.8) | | 2064 | 4.8 (0.9) | |
| | Rural | 2793 | 3.0 (1.0) | | 2291 | 2.0 (1.0) | |
| Country | Argentina | 4230 | 3.0 (0.8–4.0) | 0.44 | 2017 | 1.9 (0.8) | 0.44 |
| | Brazil | 4523 | 2.0 (0.9–8.6) | | 3030 | 1.4 (0.6–5.8) | |
| | Chile | 2919 | 7.7 (2.7–13.4) | | 2919 | 7.7 (2.7–13.4) | |
| | Colombia | 6000 | 4.0 (1.9–7.4) | | 6000 | 4.0 (1.9–7.4) | |
| | Peru | 586 | 0.3 | | 375 | 1.6 | |
| | Venezuela | 1012 | 4 | | 922 | 4 | |
| | | **Low HDL by any definition** | | | **Low HDL by HDL ≤40 mg/dL** | | |
| Variables | | N (%) | Prevalence %, median (IQR) | p value | N (%) | Prevalence %, median (IQR) | p value |
| Overall | | 85802 | 29.6 (17.7–36.9) | | 15181 | 22.6 (19.2–36.7) | |
| Setting | Urban | 13482 | 28.8 (19.5–36.4) | 0.27 | 8259 | 22.1 (19.3–36.1) | 0.14 |
| | Mixed | 67044 | 37.0 (14.1–69.5) | | 1983 | 40.1 (37.0) | |
| | Rural | 5276 | 16.4 (7.9–32.5) | | 4939 | 24.2 (12.6–32.7) | |
| Country | Argentina | 5184 | 19.6 (11.7–32.3) | 0.62 | 3866 | 25.9 (19.5–32.7) | 0.82 |
| | Brazil | 71797 | 29.6 (17.0–35.8) | | 5015 | 35.5 (13.2–39.9) | |
| | Chile | 1210 | 49.7 (22.6) | | 208 | 22.6 | |
| | Colombia | 6000 | 22.1 (15.3–56.0) | | 5506 | 20.6 (13.3–53.8) | |
| | Peru | 586 | 37 | | 586 | 37 | |
| | Venezuela | 1003 | 59.7 (9.0) | | | | |
| | | **Elevated Triglycerides by any definition** | | | **Elevated Triglycerides defined as ≥100 mg/dL for ages 2–9 and ≥130 mg/dL for ages 10–19** | | |
| Variables | | N (%) | Prevalence %, median (IQR) | p value | N (%) | Prevalence %, median (IQR) | p value |
| Overall | | 84384 | 18.1 (11.6–34.9) | | 68166 | 19.1 (10.4–26.2) | |
| Setting | Urban | 13336 | 19.4 (12.7–31.3) | 0.76 | 1611 | 20.4 (11.0–25.1) | 0.62 |
| | Mixed | 66677 | 12.3 (8.3–54.2) | | 63927 | 44.9 (12.2) | |
| | Rural | 4371 | 17.7 (10.3–42.5) | | 2628 | 10.4 (10.4) | |
| Country | Argentina | 4290 | 12.4 (11.3–19.0) | 0.51 | 1346 | 17.7 (10.4) | 0.85 |
| | Brazil | 71273 | 17.8 (10.9–36.7) | | 64836 | 17.8 (8.2–53.6) | |
| | Chile | 1210 | 20.7 (7.8) | | 208 | 15.4 (10.4) | |
| | Colombia | 6000 | 20.4 (12.3–32.4) | | 1282 | 10.4 | |
| | Peru | 586 | 46.4 | | | | |
| | Venezuela | 1003 | 26.1 (11.3) | | | | |

(Continued)

**Table 4.** (Continued)

| Variables | | Obesity by any definition | | | Obesity by BMI, WH, WL ≥95%* | | |
|---|---|---|---|---|---|---|---|
| | | N (%) | Prevalence %, median (IQR) | p value | N (%) | Prevalence %, median (IQR) | p value |
| Variables | | **High blood Pressure by any definition** | | | **High Blood Pressure by BP ≥90% for age, gender, and height** | | |
| | | N (%) | Prevalence %, median (IQR) | p value | N (%) | Prevalence %, median (IQR) | p value |
| Overall | | 21055 | 8.6 (4.6–20.6) | | 12784 | 8.6 (4.2–20.0) | |
| Setting | Urban | 6957 | 8.7 (6.0–22.0) | 0.73 | 5818 | 8.7 (6.0–29.3) | 0.67 |
| | Mixed | 10938 | 9.3 (3.6–22.1) | | 5313 | 7.6 (2.0–36.9) | |
| | Rural | 2372 | 8.0 (2.5–15.0) | | 1653 | 8.0 (4.2) | |
| Country | Argentina | 7864 | 8.6 (4.2–20.5) | 0.09 | 4215 | 8.0 (4.1–14.3) | 0.19 |
| | Brazil | 5991 | 14.3 (8.0–25.4) | | 3461 | 15.9 (11.3) | |
| | Chile | 3076 | 7.6 (5.3–32.2) | | 2286 | 7.8 (4.3–38.9) | |
| | Colombia | 2270 | 2.6 (0.7) | | 988 | 3.8 (2.6) | |
| | Ecuador | 335 | 25.1 (9.3) | | 335 | 25.1 (9.3) | |
| | Paraguay | 132 | 1.5 | | | | |
| | Peru | 586 | 0.8 | | 586 | 0.8 | |
| | Venezuela | 913 | 0.9 | | 913 | 0.9 | |

IQR = inter-quartile range; p-values calculated to 95% significance; BMI = body mass index; WH = waist to height ratio; WL = weight to length ratio; HDL = high density lipoprotein.

(waist-to-length ratio) ≥95% for age and gender (53 studies, 33.3%). In total by any definition, a median of 12.2% (Interquartile range (IQR) 6.35–17.2) children were obese with 23.9% (IQR 16.6–39.0) with overweight and/or obesity without distinction, and a median of 21.9% (IQR 10.0–33.2) specifically experiencing abdominal obesity. Eight studies (5.0%) included indigenous populations; obesity by any definition was lower in indigenous children than non-indigenous children (median, IQR: 2.5%, 1.6–5.6 vs. 12.7%, 6.5–17.5, p = 0.009). Obesity by any definition varied significantly by country, with Chile exhibiting the highest prevalence of both obesity and obesity and/or overweight without distinction (median, IQR obesity in Chile: 20.6%, 15.6–27.8 and obesity/overweight: 47.5%, 41.1–60.8) when compared to other countries (Table 4, Fig 2). Obesity by any definition was lower in rural populations than in urban or mixed populations (median, IQR obesity in rural setting: 5.6%, 1.6–12.0; in urban setting: 12.6%, 6.3–17.3; and mixed setting: 12.9%, 6.5–16.2; p = 0.08).

When looking only at obesity as defined by BMI, WH or WL ≥95%, this was higher in non-indigenous populations than indigenous populations (median, IQR for non-indigenous: 13.9%, 7.7–21.1 vs. for indigenous: 3.4%, 1.6, n = 2; p = 0.07) and differed by setting and country (Table 4). Rural populations experienced lower rates of obesity by this definition than did mixed and urban populations, which were almost equivalent (median, IQR: rural 4.9%, 2.0–9.1 vs. mixed 13.6%, 7.2–17.6 vs. urban 13.9%, 6.8–20.4, p = 0.08). These differences held for those studies in which obesity was indistinguishable from overweight (Table 4).

## Elevated waist circumference

Elevated waist circumference was reported in 43 studies (24.0% of total studies included), encompassing 38,250 individuals. The most commonly used definition of waist circumference was WC ≥90% for age, height and gender (29 studies, 67.4%). Brazil had the highest prevalence of elevated waist circumference by any definition (median, IQR: 29.6%, 10.5–48.0, p = 0.04). There were no differences in median prevalence rates of elevated waist

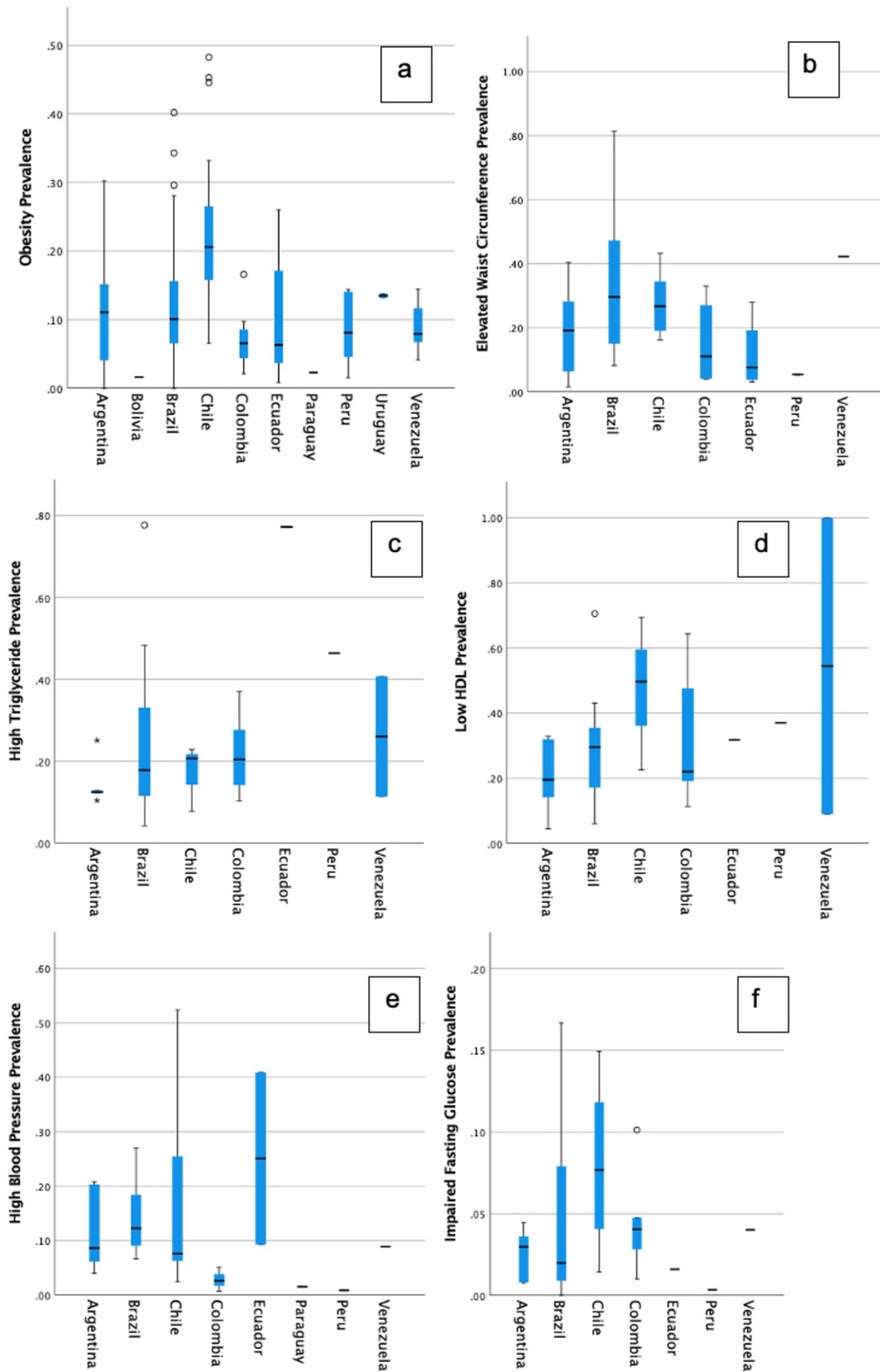

**Fig 2. Cardiometabolic risk factor prevalence in South American children by country.** Box-and-whisker plots showing median and IQR values for each CMRF by country by any definition. Y-axis is prevalence, x-axis is country. A) obesity, B) elevated waist circumference, C) high triglyceride, D) low HDL, E) high blood pressure, F) impaired fasting glucose. HDL = high-density lipoprotein.

circumference defined by WC ≥90% for age and gender between setting nor by country (Argentina, Brazil, Chile, Colombia, and Ecuador; Table 4). There was a trend to higher median prevalence in Chile and a lower prevalence in Colombia and Peru (median, IQR for Chile: 26.0%, 16.2 n = 2 vs. Colombia: 5.6%, 4.1–22.0 vs. Peru: 5.4%, 5.2% n = 2) as detailed in Fig 2.

### Impaired fasting glucose

Twenty-seven studies (15.1% of included studies) measured impaired fasting glucose and included 19,810 individuals using 4 different definitions (Table 3). The most common definition was fasting glucose ≥100 mg/dL (21, 77.8%). The median prevalence of impaired fasting glucose by any definition was 3.0% (IQR 1.0–6.6), which was comparable to the median prevalence of impaired fasting glucose by the most commonly used definition (2.8% IQR 1.0–6.7). For both impaired fasting glucose overall and studies that used the most common definition, studies with mixed settings experienced slightly higher prevalence rates over studies with strictly urban and rural settings (Table 4). Chile trended to a higher median prevalence of impaired fasting glucose than other countries in the analysis, but this did not reach statistical significance (Table 4).

We also recorded the prevalence of insulin resistance and hyperinsulinemia in studies that reported these risk factors. The median prevalence rates of hyperinsulinemia and insulin resistance by any definition was 6.3% (IQR 3.3–12.7) and 5.4% (IQR 4.8–27.3), respectively. The number of studies that reported on these risk factors (n = 7, 3.9% of included studies) for hyperinsulinemia, n = 9 (5.0%) for insulin resistance) was too small to complete summary statistics on this data.

### High blood pressure

Thirty-three studies measured high blood pressure (18.5% of included studies) encompassing 21,055 individuals and 8 different definitions (Table 3). The most common definition of HBP was SBP or DBP ≥90% for age, height and gender (22, 66.7%). The median prevalence of high blood pressure by any definition was 8.6% (IQR 5.1–20.6), and the median prevalence as defined by BP ≥90% for age, gender and height was 8.6% (IQR 4.2–20.0). HBP did not differ greatly between study setting or country, although Ecuador exhibited a higher trend in HBP than other countries (median, IQR 25.1%, 9.3 n = 2). Prevalence rates are detailed in Table 4.

### Dyslipidemia

Thirty-eight studies (21.2% of studies included) measured low HDL including 85,802 individuals from 6 countries using 9 different definitions (Table 3). Thirty studies (16.8% of included studies) measured elevated triglycerides including 84,384 individuals from 6 countries using 12 different definitions. Data is summarized in Table 4.

**Elevated triglycerides.** Less than 20% of South American children experienced elevated triglycerides (TG) (median, IQR 18.1%, 11.6–34.9) by any definition; this was similar to the prevalence as defined by the most common definitions of TG ≥100 mg/dL in children age 2–9 years together with TG ≥130 mg/dL in children 10–19 years (median, IQR 19.1%, 10.4–26.2). There were no notable differences in hypertriglyceridemia between children in different study settings, though children in mixed settings in studies that used the two most common definitions had slightly higher levels of TG than children in strictly urban and rural settings (median, IQR in mixed: 44.9%, 12.2 n = 2; vs. urban: 20.4%, 11.0–25.1 vs. rural: 10.4%, n = 1).

Venezuelan children had slightly higher prevalence of elevated TG than children in other countries, but this did not reach statistical significance (median, IQR 26.1%, 11.3 n = 2, Table 4).

**Low HDL cholesterol.** The most commonly used definition of low HDL was HDL≤40 mg/dL (18 studies, 47.4% of studies reporting on dyslipidemia; Table 3). The median prevalence of low HDL cholesterol by any definition was 29.6% (IQR 17.7–36.9) and 22.6% (IQR 19.2–36.7) as defined by HDL ≤40 mg/dL. Low HDL did not vary greatly by either study setting or country for all studies or for those that used the most common definition (HDL ≤40 mg/dL) (see Table 4). Rural populations experienced a slightly lower prevalence of low HDL in all studies (median, IQR 16.4%, 7.9–32.5) versus mixed and urban populations (median, IQR mixed: 37.0%, 14.4–69.5 vs. urban: 28.8%, 19.5–36).

## Discussion

### Comparison to other literature

Our study provides the most comprehensive assessment of CMRF in South American children. Krishnan and Short [191] assessed CMRF prevalence in South American children with type 1 diabetes mellitus, but we did not find any other articles that focused on the general South American child and adolescent population. In comparison to other geographic populations, CMRF prevalence in South American children, and children in general, is poorly characterized. We could find few meta-analyses that focus on regional burdens of CMRF prevalence; many focus on a subset of the child and adolescent population, such as Krishnan and Short, or are concerned with assessing screening tools [191]. Most regionally-focused literature on CMRF in children and adolescents details obesity and overweight prevalence, while dyslipidemia and impaired glucose metabolism are minimally characterized. The World Health Organization estimates that as of 2016, roughly 12.8% of children and adolescents aged 10–19 in the Americas experience obesity, the highest of all regions [192]. The next-closest region is the Western Pacific, with 7.4% of children experiencing obesity, which demonstrates how alarmingly high the prevalence of childhood and adolescent obesity is in the Americas as compared to the rest of the globe and the need to focus additional attention in this vulnerable region. In addition, we chose to focus the scope of this article on South America. In our preliminary investigation of this topic, we found a paucity of information on CMRF in Caribbean children.

### Controversy of metabolic syndrome in children

Although the utility and consistency of a diagnosis of MetS in children is controversial, a more comprehensive screening tool for population-level prevalence of cardiometabolic health factors does not yet exist. Many organizations and publications, including the AHA, have questioned the utility of diagnosing MetS in children. Goodman et al, 2007 showed that potentially large proportions of children diagnosed with MetS do not meet criteria upon follow-up 3 to 6 years later [193]. On the other hand, there is literature to support an increased risk of persistence of MetS into adulthood [194]. In one cohort study of 771 adults (mean age 38 years) who had previously participated in the Lipid Research Clinics study as children and adolescents, the incidence of self-reported cardiovascular disease was more common in adults who exhibited metabolic syndrome traits as children than in those who did not (19.4 versus 1.5%, odds ratio 14.6, 95% CI 4.8–45.3) [195]. Of 31 children who had metabolic syndrome traits as children, 21 (68%) had metabolic syndrome as adults. In the context of these results, the high burden of obesity and dyslipidemia demonstrated in our analysis suggests a future rise in CMRF in the South American adult population.

## Limitations

The greatest limitation in interpreting this data is the heterogeneity of definitions used for individual CMRF. As displayed in Table 2, there were 4–15 definitions used for each risk factor. We addressed this issue by generating additional summary statistics with data that shared the same cutoff definition. This issue would be avoided in future studies if there was an international consensus on and wider acceptance of standardized definitions for CMRF in children.

We could not find any published data from Guyana, Suriname, French Guiana nor the Falkland Islands. The absence of data from these countries and territories limits the generalizability of this study to the whole region. Additionally, we extracted as much demographic information as possible; however, we were unable to extract basic demographic data such as gender for each definition, as many studies reported more than one risk factor but not necessarily gender distribution or ages per risk factor. Therefore, we do not know if CMRF are clustered more in one gender over another or one age group over another. A further limitation was the minimal inclusion of indigenous or other minority populations. Although many South American populations include indigenous members, only a handful of studies specifically noted the inclusion of majority indigenous groups and no studies specifically reported on other minority groups. Our analysis therefore likely underestimates the prevalence of CMRF in these more vulnerable individuals.

Additionally, when compiling the titles from the basic Google Scholar search, there was a server error after uploading 980 titles from this database despite multiple attempts to upload all results. Therefore, only 980 of the 1,060 titles from this search were able to be included for screening. It is unknown how many of the eighty titles would have met our inclusion criteria. Titles from Google Scholar were imported using the Mendeley plug-in for Google Chrome. This software did not differentiate between Spanish and English versions of the same text, so there were many duplicate titles that were imported and had to be screened out. This led to a high number of duplicate titles.

## Collective trends

Of all CMRF, obesity was the most widely described and thus generated the largest differences between variables (ex. study setting and indigenous populations). This is likely due to sample size, and it is possible that the variables evaluated for other CMRF are underpowered. Low HDL and overweight and/or obese status were the two most prevalent CMRF, and impaired fasting glucose and HBP were the lowest. These trends are consistent with the results of Li et al., who found similar rates of CMRF in severely obese children in the United States (HBP: 9.9%, low HDL: 40%, high TG: 30.0%) [196]. The higher rates of obesity and overweight are not surprising, though the high prevalence of HDL was not expected. This finding could be influenced by the growing urbanization of South America resulting in increased consumption of processed foods and sedentary lifestyle of South American children in mixed and urban areas [197].

Brazil, Chile, Colombia, and Argentina were the most represented in terms of number of studies. Though Peru had the largest population included in our analysis, this is due to one study (Torres-Roman et al.) including a population of over 2 million. Our study highlights an especially high prevalence of many CMRF in Chilean children.

Overall, children in rural settings and indigenous children trended to lower prevalence rates of CMRF. This differs from other studies on the nutritional status of indigenous populations worldwide, which document a growing double burden of undernutrition and obesity in indigenous children [198]. This inconsistency could be attributed to the small sample sizes and lack of available data on indigenous populations in South America.

### Knowledge gaps

Many South American countries were underrepresented or not represented at all in our analysis due to a lack of available data. There were three or fewer studies that contained data from Bolivia, Paraguay, and Uruguay for each CMRF, and many countries only had data reported on obesity. Many countries, including French Guiana and Suriname, did not have any available or relevant articles. As outlined above, indigenous and minority populations should be the focus of future studies, as they were vastly underrepresented in our comprehensive analysis.

## Conclusion

The high prevalence of obesity, overweight, and dyslipidemia in South American children are concerning. This study provides strong evidence to policy makers and healthcare workers to design interventions to address CMRF clustering in children to curb future disease burden. Our study also denotes the analytical difficulties that arise from a lack of standardized and widely accepted CMRF definitions in children. Additionally, future studies should focus on characterizing CMRF in underrepresented countries and populations, delineating the burden distribution by gender and age, and confirming the finding here of higher clustering of CMRF in urban populations. It is our hope that this meta-analysis will stimulate additional research to stem the rapid regional and global rise in childhood metabolic syndrome and early cardiovascular disease.

## Supporting information

**S1 Checklist. PRISMA 2020 checklist.**
(DOCX)

## Author Contributions

**Conceptualization:** Carolyn M. H. Singleton, Sumeer Brar, Nicole Robertson, George J. Fuchs, III, Aric Schadler, Suzanna L. Attia.

**Data curation:** Carolyn M. H. Singleton, Sumeer Brar, Nicole Robertson, Lauren DiTommaso, Aric Schadler, Suzanna L. Attia.

**Formal analysis:** Carolyn M. H. Singleton, Sumeer Brar, Nicole Robertson, Lauren DiTommaso, George J. Fuchs, III, Aric Schadler, Suzanna L. Attia.

**Investigation:** Carolyn M. H. Singleton, Sumeer Brar, Suzanna L. Attia.

**Methodology:** Carolyn M. H. Singleton, Sumeer Brar, Nicole Robertson, Lauren DiTommaso, Aric Schadler, Aurelia Radulescu, Suzanna L. Attia.

**Project administration:** Suzanna L. Attia.

**Resources:** Carolyn M. H. Singleton, Nicole Robertson, George J. Fuchs, III, Suzanna L. Attia.

**Supervision:** George J. Fuchs, III, Aurelia Radulescu, Suzanna L. Attia.

**Validation:** Sumeer Brar, Nicole Robertson, Lauren DiTommaso, Aurelia Radulescu, Suzanna L. Attia.

**Visualization:** Carolyn M. H. Singleton, Sumeer Brar, Suzanna L. Attia.

**Writing – original draft:** Carolyn M. H. Singleton, Aric Schadler, Suzanna L. Attia.

**Writing – review & editing:** Carolyn M. H. Singleton, Suzanna L. Attia.

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
