## [Decision Letter · Decision Letter 0]

24 Apr 2023

PONE-D-22-19334Cardiometabolic Risk Factors in South American Children: A Systematic Review and Meta-AnalysisPLOS ONE

Dear Dr. Haugh,

Thank you for submitting your manuscript to PLOS ONE. I sincerely apologise for the unusually delayed review timeframe. Your manuscript has been assessed by one reviewer, whose comments are appended below. After careful consideration, we feel that it has merit but does not fully meet PLOS ONE’s publication criteria as it currently stands. Therefore, we invite you to submit a revised version of the manuscript that addresses the points raised during the review process. Please note that we have only been able to secure a single reviewer to assess your manuscript. We are issuing a decision on your manuscript at this point to prevent further delays in the evaluation of your manuscript. Please be aware that the editor who handles your revised manuscript might find it necessary to invite additional reviewers to assess this work once the revised manuscript is submitted. However, we will aim to proceed on the basis of this single review if possible.

We look forward to receiving your revised manuscript.

Kind regards,

Emily Chenette

Editor in Chief

PLOS ONE

Journal Requirements:

2. Please ensure that your search is up to date and any relevant studies published since 2021 are included in your systematic review.

Reviewers' comments:

Reviewer's Responses to Questions

**Comments to the Author**

1. Is the manuscript technically sound, and do the data support the conclusions?

Reviewer #1: Yes

2. Has the statistical analysis been performed appropriately and rigorously? 

Reviewer #1: Yes

3. Have the authors made all data underlying the findings in their manuscript fully available?

Reviewer #1: Yes

4. Is the manuscript presented in an intelligible fashion and written in standard English?

Reviewer #1: Yes

5. Review Comments to the Author

Reviewer #1: I am very grateful for the opportunity to review this manuscript. The manuscript aims to study the cardiometabolic risk factors in South American children through a systematic review and meta-analysis.

A great effort was done in the preparation of this manuscript and the data presented might give a valuable insight about the high rates of obesity, overweight and dyslipidemia in these children. In addition, it highlights the knowledge gap and gives recommendation for the possible future research in this field.

I have only few comments:

Methods

In the search strategy section, database search date was not mentioned.

Line 164: “priams”, should be Prisma

Line 173: “baseline data was recorded”, should be “were”

Results

Regarding the definitions used for cardiometabolic risk factors, some definitions are exactly the same but using different units. For example triglycerides 150 mg/dl is the same as 1.96 mmol/L and blood glucose of 100 mg/dl is almost the same as 5.55 mmol/l. I would rather merge these similar definitions into one rather than considering them as two different definitions.

In the study that used the cut off value of “Fasting glucose ≥126 mg/dL”, was that used to define impaired glucose tolerance/prediabetes or for diabetes mellitus?

Line 295: “data is”, should be “are”

Discrepancies between the text and tables

Line 227: IQR: 6.35-17.2. In the table: 6.3-17.3

Line 228 IQR (16.6-39.0). In the table: 16.5-39.3

Line 229: median IQR 21.9% (IQR 10.0-33.2). In the table: 21.9 (9.4-33.9) -

Lines 236-237 urban setting: 12.6%, 6.3-17.3. In the table: 12.6 (6.5-18.0)

Line 250 P-value is 0.07. In the table: 0.007

Lines 257-258: "the most commonly used definition of waist circumference was WC ≥90% for age, height and gender" Lines 260-261: " elevated waist circumference defined by WC ≥90% for age and gender Table 3:" WC ≥90% for age, height and gender" Table 4: "WC ≥90% for age and gender"

Line 286: (IQR 5.1-20.6). In table 4: (4.6-20.6)

Line 303: "IQR in mixed: 4.49%. In table 4: 44.9

Line314: "mixed: 37.0%, 14.4-69.5". In table 4: 14.1-69.5) - urban: 28.8%, 19.5-36. In table 4: 28.8 (19.5-36.4)

Tables

In the columns heading it is written N(%), it is better to remove the “%” as no percentages where mentioned in these columns

** In both text and table, data that are not represented as median (IQR), has to be explained to the readers

6. PLOS authors have the option to publish the peer review history of their article (what does this mean?). If published, this will include your full peer review and any attached files.

Reviewer #1: No

---

## [Author Response · Author response to Decision Letter 0]

11 Jun 2023

Methods

1. In the search strategy section, database search date was not mentioned.

Response: Thank you for bringing up this limitation in our methods. Unfortunately, though we did record dates of searches for some database searches, we did not record all of them. For this reason, we did not include the dates of the searches.

2. Line 164: “priams”, should be Prisma

Response: This change was made.

3. Line 173: “baseline data was recorded”, should be “were”

Response: This was addressed.

Results

1. Regarding the definitions used for cardiometabolic risk factors, some definitions are exactly the same but using different units. For example triglycerides 150 mg/dl is the same as 1.96 mmol/L and blood glucose of 100 mg/dl is almost the same as 5.55 mmol/l. I would rather merge these similar definitions into one rather than considering them as two different definitions.

Response: Thank you for suggesting this way to streamline the presentation of our findings. We have merged the similar definitions and updated Table 3.

2. In the study that used the cut off value of “Fasting glucose ≥126 mg/dL”, was that used to define impaired glucose tolerance/prediabetes or for diabetes mellitus? 

Response: We reviewed that article closer, and it did indeed use that definition to defint diabetes mellitus but did not provide a definition for impaired fasting glucose. Since this study did not provide relevant data on that particular risk factor based on our inclusion criteria, we removed the data from that study for that risk factor.

-Line 295: “data is”, should be “are”

Response: Thank you, this has been addressed. 

Discrepancies between the text and tables

3. Line 227: IQR: 6.35-17.2. In the table: 6.3-17.3

Response: Thank you, the text has been changed to agree with the correct data in the table.

4. Line 228 IQR (16.6-39.0). In the table: 16.5-39.3

Response: Thank you, the text has been changed to agree with the correct data in the table.

5. Line 229: median IQR 21.9% (IQR 10.0-33.2). In the table: 21.9 (9.4-33.9)

Response: Thank you, the text has been changed to agree with the correct data in the table.

6. Lines 236-237 urban setting: 12.6%, 6.3-17.3. In the table: 12.6 (6.5-18.0)

Response: Thank you, the text has been changed to agree with the correct data in the table.

7. Line 250 P-value is 0.07. In the table: 0.007

Response: Thank you, the text has been changed to agree with the correct data in the table. 

8. Lines 257-258: "the most commonly used definition of waist circumference was WC ≥90% for age, height and gender" Lines 260-261: " elevated waist circumference defined by WC ≥90% for age and gender Table 3:" WC ≥90% for age, height and gender" Table 4: "WC ≥90% for age and gender"

Response: Updated definition in lines 260-261 to include full definition.

9. Line 286: (IQR 5.1-20.6). In table 4: (4.6-20.6)

Response: Thank you, the text has been changed to agree with the correct data in the table. 

 10. Line 303: "IQR in mixed: 4.49%. In table 4: 44.9

Response: Thank you, the text has been changed to agree with the correct data in the table. 

11. Line314: "mixed: 37.0%, 14.4-69.5". In table 4: 14.1-69.5) - urban: 28.8%, 19.5-36. In table 4: 28.8 (19.5-36.4)

Response: Thank you, the text has been changed to agree with the correct data in the table.

Tables

1. In the columns heading it is written N(%), it is better to remove the “%” as no percentages where mentioned in these columns

Response: Thank you for this comment, we have removed the “%” 

2. In both text and table, data that are not represented as median (IQR), has to be explained to the readers

Response: In tables, inserted * with explanation in caption of “*Unable to provide full IQR as study N=2” and ** with caption of “**Unable to provide IQR as study N=1.” In the text, inserted “Study N=1” or “Study N=2” as appropriate.

---

## [Decision Letter · Decision Letter 1]

16 Aug 2023

PONE-D-22-19334R1Cardiometabolic Risk Factors in South American Children: A Systematic Review and Meta-AnalysisPLOS ONE

Dear Dr. Attia,

Thank you for submitting your manuscript to PLOS ONE. After careful consideration, we feel that it has merit but does not fully meet PLOS ONE’s publication criteria as it currently stands. Therefore, we invite you to submit a revised version of the manuscript that addresses the points raised during the review process.

There are some minor comments that need to be addressed before the manuscript could be published 

We look forward to receiving your revised manuscript.

Kind regards,

Patricia Khashayar

Academic Editor

PLOS ONE

Journal Requirements:

Reviewers' comments:

Reviewer's Responses to Questions

**Comments to the Author**

1. If the authors have adequately addressed your comments raised in a previous round of review and you feel that this manuscript is now acceptable for publication, you may indicate that here to bypass the “Comments to the Author” section, enter your conflict of interest statement in the “Confidential to Editor” section, and submit your "Accept" recommendation.

Reviewer #2: (No Response)

2. Is the manuscript technically sound, and do the data support the conclusions?

Reviewer #2: Yes

3. Has the statistical analysis been performed appropriately and rigorously? 

Reviewer #2: Yes

4. Have the authors made all data underlying the findings in their manuscript fully available?

Reviewer #2: (No Response)

5. Is the manuscript presented in an intelligible fashion and written in standard English?

Reviewer #2: Yes

6. Review Comments to the Author

Reviewer #2: I reviewed the manuscript entitled “Cardiometabolic Risk Factors in South American Children: A Systematic Review and Meta-Analysis”. This is an interesting article assessing the effect of WWI on stroke risk. The manuscript is in the scope of the journal, but certain shortcomings should be addressed before the article could be published:

• Several language and grammar editing is required.

• The introduction section is too long. Some information mentioned here belongs to the discussion section

• Why did you exclude Caribbean islands from geographical region of South America? Moreover, as mentioned in the article the data from several South American countries data are missing, how does this affect the generalizability of the results to South America?

• In the abstract you have mentioned 1,540 screened studies, however, in the results (Line 206) “2,181 studies screened” is written. Which one is the correct number?

• Figure 2 is missing in the manuscript.

• As you already mentioned in the limitation, heterogeneity of definitions of CMRF is the main concern. How have you addressed that. Please elaborate.

• Is the inter-study heterogeneity studied?

7. PLOS authors have the option to publish the peer review history of their article (what does this mean?). If published, this will include your full peer review and any attached files.

Reviewer #2: **Yes: **Pouria Khashayar

---

## [Author Response · Author response to Decision Letter 1]

28 Sep 2023

Thank you for your constructive comments and the time you took to review our work. Please find our responses below. Please note that the line references are given as lines in the tracked changes version/ lines in clean version.

1. We have reviewed the document for language and grammar edits.

2. We made several edits to the Introduction section, as detailed in the revised manuscript. Notably, we moved two paragraphs about the utility of metabolic syndrome in children to the Discussion section.

3. We excluded the Caribbean Islands due to their increased cultural and culinary heterogeneity as compared to the countries conventionally considered South America. We have addressed this in lines 136-137/112-113. In results and discussion (lines 212-214/187-188 and 372-373/340-341 respectively), we list the South American countries and territories that did not have data. We discussed this as a limitation to generalizability in our Discussion section (lines 373-374 / 341-342).

4. The number originally listed in the abstract was the total number of articles screened before we added the additional, targeted Google searches. This number has been updated to match the number in the body of the article.

5. Figures 1 and 2 are in a separate document per editor’s instructions. Please let the editor know if you are unable to view this document.

6. We address heterogeneity in our analysis by completing separate statistical analysis by the most used definition of each CMRF to assess the effect of the differing definitions on our outcomes (addressed in Methods 201-203/175-177 and Discussion in lines 366-371/335-336. We did not find a statistical test that would be appropriate to test the heterogeneity of our studies.

---

## [Decision Letter · Decision Letter 2]

23 Oct 2023

Cardiometabolic Risk Factors in South American Children: A Systematic Review and Meta-Analysis

PONE-D-22-19334R2

Dear Dr. Attia,

We’re pleased to inform you that your manuscript has been judged scientifically suitable for publication and will be formally accepted for publication once it meets all outstanding technical requirements.

Kind regards,

Patricia Khashayar

Academic Editor

PLOS ONE

Additional Editor Comments (optional):

Reviewers' comments:

Reviewer's Responses to Questions

**Comments to the Author**

1. If the authors have adequately addressed your comments raised in a previous round of review and you feel that this manuscript is now acceptable for publication, you may indicate that here to bypass the “Comments to the Author” section, enter your conflict of interest statement in the “Confidential to Editor” section, and submit your "Accept" recommendation.

Reviewer #2: All comments have been addressed

2. Is the manuscript technically sound, and do the data support the conclusions?

Reviewer #2: Yes

3. Has the statistical analysis been performed appropriately and rigorously? 

Reviewer #2: Yes

4. Have the authors made all data underlying the findings in their manuscript fully available?

Reviewer #2: Yes

5. Is the manuscript presented in an intelligible fashion and written in standard English?

Reviewer #2: Yes

6. Review Comments to the Author

Reviewer #2: (No Response)

7. PLOS authors have the option to publish the peer review history of their article (what does this mean?). If published, this will include your full peer review and any attached files.

Reviewer #2: **Yes: **Pouria Khashayar

---

## [Editor Report · Acceptance letter]

13 Nov 2023

PONE-D-22-19334R2 

Cardiometabolic Risk Factors in South American Children: A Systematic Review and Meta-Analysis 

Dear Dr. Attia:

I'm pleased to inform you that your manuscript has been deemed suitable for publication in PLOS ONE. Congratulations! Your manuscript is now with our production department. 

Kind regards, 

on behalf of

Dr. Patricia Khashayar 

Academic Editor

PLOS ONE